# Examining Disparities in Current E-Cigarette Use among U.S. Adults before and after the WHO Declaration of the COVID-19 Pandemic in March 2020

**DOI:** 10.3390/ijerph20095649

**Published:** 2023-04-26

**Authors:** Hadii M. Mamudu, David Adzrago, Oluwabunmi Dada, Emmanuel A. Odame, Manik Ahuja, Manul Awasthi, Florence M. Weierbach, Faustine Williams, David W. Stewart, Timir K. Paul

**Affiliations:** 1College of Public Health, East Tennessee State University, Johnson City, TN 37614, USA; ahujam@etsu.edu (M.A.); awasthim@etsu.edu (M.A.); 2Center for Cardiovascular Risk Research, College of Public Health, East Tennessee State University, Johnson City, TN 37614, USA; weierbach@etsu.edu (F.M.W.); stewardw@etsu.edu (D.W.S.); tpaul5@uthsc.edu (T.K.P.); 3Center for Health Promotion and Prevention Research and School of Public Health, The University of Texas Health Science Center at Houston, Houston, TX 77030, USA; david.adzrago@uth.tmc.edu; 4Department of Occupational Safety and Health, Murray State University, 157 Industry and Technology Center, Murray, KY 42071, USA; odada1@murraystate.edu; 5Department of Environmental Health Sciences, Ryals Public Health Building (RPHB), University of Alabama at Birmingham, 1665 University Boulevard, Birmingham, AL 35233, USA; david19@uab.edu; 6College of Nursing, East Tennessee State University, Johnson City, TN 37614, USA; 7Division of Intramural Research, National Institute on Minority Health and Health Disparities, National Institutes of Health, Bethesda, MD 20892, USA; faustine.williams@nih.gov; 8Bill Gatton College of Pharmacy, East Tennessee State University, Johnson City, TN 37614, USA; 9Division of Medicine, University of Tennessee at Nashville/Ascension Saint Thomas Hospital, Nashville, TN 37205, USA

**Keywords:** e-cigarettes, COVID-19 pandemic, disparities, sexual identities, health conditions, substance use

## Abstract

This paper aims to estimate the prevalence of e-cigarette use before and after the COVID-19 pandemic declaration and to delineate disparities in use across subpopulations. Data were derived from the 2020 Health Information National Trends Survey (*N* = 3865) to conduct weighted multivariable logistic regression and marginal analyses. The overall prevalence of current e-cigarette use increased from 4.79% to 8.63% after the COVID-19 pandemic declaration. Furthermore, non-Hispanic Black people and Hispanic people had lower odds of current e-cigarette use than non-Hispanic White people, but no significant differences were observed between groups before the pandemic. Compared to heterosexual participants, sexual minority (SM) participants had higher odds of current e-cigarette use after the declaration, with insignificant differences before. People who had cardiovascular disease conditions, relative to those without, had higher odds of current e-cigarette use after the declaration, but no group differences were found before the declaration. The marginal analyses showed that before and after the pandemic declaration, SM individuals had a significantly higher probability of using e-cigarettes compared to heterosexual individuals. These findings suggest the importance of adopting a subpopulation approach to understand and develop initiatives to address substance use, such as e-cigarettes, during pandemics and other public health emergencies.

## 1. Introduction

As the cases and fatalities due to the novel coronavirus disease 2019 (COVID-19) [1,2,3] increased worldwide, including the United States (U.S.), the World Health Organization (WHO) declared COVID-19 a global pandemic on 11 March 2020 [2]. Before this Declaration, electronic cigarette (e-cigarette) use had spread among varying demographic groups in the U.S. [4,5,6]. The use of e-cigarettes is a major public health concern because of the short-term adverse health consequences affecting the pulmonary and cardiovascular systems [7] that have been reported with concerns about the possibility of long-term effects [8,9,10,11,12]. While major pulmonary complications from COVID-19 are well documented [13], there is conflicting evidence on how smoking and/or vaping aggravates the symptoms [14,15,16]. Studies have reported that people who smoke, compared to those who do not smoke, may be more likely to experience severe respiratory-related diseases from COVID-19 [17,18,19]. In contrast, as of May 2020, an ecological study in 38 European countries reported a negative association between smoking prevalence and COVID-19 incidence [20]. Conversely, according to the WHO and U.S. Centers for Disease Control and Prevention (CDC), smoking is a risk factor for severe illnesses from COVID-19 [21,22]; however, there is no affirmation on the association between e-cigarette use and COVID-19 risk, except in young adults [23], serving as the impetus for this study.

The WHO Declaration of the COVID-19 pandemic culminated in most countries, including the U.S., enacting mitigation measures such as lockdown orders that prevented human interaction to avert the spread of the virus [3]. These restrictions had the potential to affect access to e-cigarettes, although this has not been ascertained. Before the pandemic, approximately 8.1 million U.S. adults were e-cigarette users, with variations across sociodemographic characteristics; 4.3% of men were current e-cigarette users compared to 2.3% of women. This study also reported differences in e-cigarette use based on race, as 3.7% non-Hispanic white adults were current e-cigarette users, compared to 2.5% Hispanic, 1.6% non-Hispanic black, and 2.2% non-Hispanic Asian [24]. It has been reported that access to e-cigarettes is primarily via vape shops, retail stores, and online [4,23]; however, these access points were potentially altered during the pandemic due to the lockdown. There is sparse information on e-cigarette use among U.S. adults after the WHO Declaration and during the lockdown. Indeed, as of November 2022, there is a paucity of studies assessing the association between e-cigarette use and the COVID-19 pandemic declaration, and between and within population subgroups, including sex, race/ethnicity, sexual identity, rural/urban commuting area, medical/mental conditions, education, and general health status.

Studies have examined the use of tobacco products during the COVID-19 pandemic, with mixed results [25,26,27,28]. However, the impact of the COVID-19 pandemic on e-cigarette use among varying sociodemographic groups and sexual identities is yet to be fully explored, generating the critical need for such investigation to inform policy initiatives.

Several studies have investigated changes in e-cigarette use in different population subgroups during the COVID-19 pandemic, particularly among youths and young adults [23,29,30]. Some of these studies’ results were attributed to a lack of access to e-cigarettes and adult supervision [23,30]. There is, however, a different and inconsistent narrative for the adult population. One study examined changes in e-cigarette use among U.S. adults and discovered 41% and 23% of participants with no reported change and increased use, respectively [31]. In contrast, another study assessed how the lockdown order impacted the use of tobacco products, including e-cigarettes, among a sample of adults living in central California; they reported lower odds of e-cigarette users after the lockdown than before the lockdown [32]. These inconsistencies in the existing literature and the absence of studies using nationally representative data suggest the need for more studies to ascertain the disparities in e-cigarette use before and after the COVID-19 pandemic in U.S. adults.

This study aimed to use nationally representative data to estimate the prevalence of e-cigarette use before and after the COVID-19 pandemic declaration, determine the relationships between e-cigarette use among U.S. adults and the COVID-19 pandemic declaration, and delineate disparities in e-cigarette use before and after the declaration across population subgroups. This nationally representative study of the effects of COVID-19 on e-cigarette use among U.S. adults will provide critical data that could be used to inform public health and tobacco policy intervention programs during this COVID-19 pandemic era and future public health emergencies.

## 2. Methods

### 2.1. Study Design

The data for this cross-sectional survey were extracted from the 2020 Health Information National Trends Survey (HINTS 5 cycle 4) de-identified public-use file. HINTS is a large, nationally representative annual cross-sectional survey conducted among a sample of a U.S. civilian, noninstitutionalized adult population aged 18 years or older using two-stage stratified random sampling. In the first stage, a stratified sample of addresses was selected from a file of residential addresses. In the second stage, one adult was selected within each sampled household.

Detailed information about the methods and survey questions can be found in Health Information National Trends Survey 5 (HINTS 5) Cycle 4 [33] and Rutten et al. [34] Health-related information and behaviors, including e-cigarette use, physical activity, and cardiovascular diseases (CVDs), were assessed among adults in HINTS. We used the most recent HINTS data, HINTS 5, Cycle 4 (*N* = 3865), which were collected between February through June 2020. The HINTS datasets are de-identified and publicly available through the U.S. National Cancer Institute website. The Institutional Review Board of East Tennessee State University reviewed and exempted this study.

### 2.2. Measures

Current e-cigarette use status was examined as the dependent variable. This variable was derived from two survey questions: (1) “Have you ever used an e-cigarette, even one or two times?” (Yes/No) and (2) “Do you now use an e-cigarette every day; some days; or not at all?” Current e-cigarette use status was defined as never a user if the participant had never used e-cigarettes in question one and a current user if the participant reported “yes” in question one and now used it daily or somedays in question two. Those who responded “yes” in question one but “not at all” in question two, classified as former users who do not currently use, were omitted to compare the e-cigarette use behavior of only never (“no” to question one) and current (“yes” to question one and “every day or some days” to question two) users.

The independent variables were COVID-19 pandemic declaration, sexual identity, race/ethnicity, and age (18–25, 26–34, 35–49, 50–64, or 65+). The COVID-19 pandemic declaration variable, obtained from the HINTS 5 Cycle 4 methodology report, was defined as responses before and after the WHO Declaration of the COVID-19 pandemic on 11 March 2020 [33]. The HINTS 5 Cycle 4 methodology report suggests that the COVID-19 pandemic declaration variable can be used to assess the impact of COVID-19 before and after COVID-19 became a concern in the U.S. and an international pandemic [33]. Hence, survey responses received by Westat before the COVID-19 pandemic declaration were categorized as “before” the COVID-19 pandemic, and responses after the declaration were considered “after” [33]. Sexual identity was assessed by asking the participants to self-report (1 = heterosexual, or straight, 2 = homosexual, or gay or lesbian, or 3 = bisexual). Due to small cell counts or samples in the sexual identity subgroups, we dichotomized the sexual identity variable to 1 = heterosexual or straight and 2 = sexual minorities (SM; homosexual, or lesbian/gay, or bisexual). Race/ethnicity included non-Hispanic White, non-Hispanic Black/African American, Hispanic, non-Hispanic Asian, and non-Hispanic others.

The covariates were self-reported sex (male or female), level of education completed (less than High School, High School graduate, some college, or college graduate/higher), total family annual income (<$20,000, $20,000 to <$35,000, $35,000 to <$50,000, $50,000 to <$75,000, or  ≥ $75,000), employment status (employed or unemployed), health insurance status (yes or no), rural–urban commuting area (metropolitan or micropolitan/small town/rural), U.S. census region (Northeast, Midwest, South, or West), number of adult household members (less than two persons or at least two persons), general health status (excellent/very good/good or fair/poor), moderate physical activity intensity (either none or one/more days per week), and cardiovascular disease (CVD) condition (yes or no). Current anxiety/depression status was derived from Patient Health Questionnaire-4 (PHQ-4), which assesses symptoms/signs of anxiety/depression in the HINTS 5 survey. The total PHQ-4 scores range from 0–12, where scores are rated as normal/negative (0–2), mild (3–5), moderate (6–8), and severe (9–12) [35,36]. Body mass index (BMI) was used as a continuous variable (scores range from 10.9–73.8) due to limited samples in BMI subgroups.

### 2.3. Statistical Analyses

First, we calculated percentages and means/standard deviations (weighted) and frequencies (unweighted) by COVID-19 pandemic declaration categories, participants’ sociodemographic characteristics, moderate physical activity intensity, CVD status, current anxiety/depression symptoms, and current e-cigarette use status, whereas means and standard deviation were calculated for BMI. Next, we conducted three logistic regression analyses. In Model 1, we examined the association of current e-cigarette use with the COVID-19 pandemic declaration, sexual identity, race/ethnicity, and age, adjusting for sociodemographic characteristics, BMI, moderate physical activity intensity, CVD status, and current anxiety/depression symptoms. We further assessed whether the association between e-cigarette use and the COVID-19 pandemic declaration varies between and within population subgroups (gender, race/ethnicity, sexual identity, age, rural/urban commuting area, CVD, current anxiety/depression symptoms, education, and general health). We examined the interactions between the COVID-19 pandemic declaration and gender, race/ethnicity, sexual identity, age, rural/urban commuting area, CVD, current anxiety/depression symptoms, education, and general health, respectively, using joint significant effects with the Adjusted Wald test. We found significant interactions only between the COVID-19 pandemic declaration and race/ethnicity (*p* = 0.007) and sexual identity (*p* = 0.039). Additionally, we used marginal estimates/predicted values and margin plots to examine (1) the differences in current e-cigarette use between and within the COVID-19 pandemic declaration and race/ethnicity (Figure 1), and differences in current e-cigarette use between and within the COVID-19 pandemic declaration and sexual identity (Figure 2). In Model 2, we examined the association of current e-cigarette use with the covariates by responses before the COVID-19 pandemic declaration. In Model 3, we examined the Model 2 variables by responses after the COVID-19 pandemic declaration. We assessed multicollinearity and the mean variance inflation factor was 1.18, which was lower than the 10+ threshold to suggest serious collinearity. All statistical analyses were performed using STATA/SE, version 16.1 [37] and the analyses were weighted using sampling weight and replicate weight to account for the complex and stratified sampling design. All statistical analyses were performed as 2-sided with a *p*-value of 0.05, 95% confidence intervals (95% CIs), and adjusted odds ratios (AORs).

## 3. Results

### 3.1. Descriptive Statistics

Table 1 presents the sociodemographic, health behavior, and anxiety/depression characteristics of 3865 adults (representing 253,815,197 of the total U.S. adult population) before and after the WHO declared COVID-19 a pandemic. Among the participants, a weighted total of 64.69% provided responses after the COVID-19 pandemic declaration. These participants include people aged 35–49 years (27.04%), females (52.63%), non-Hispanic Whites (59.63%), had some college degree (38.92%), had a total family annual income of ≥$75,000 (40.80%), and lived in the metropolitan commuting area (88.44%). Before the declaration, the sample was comprised of people aged 50–64 years (33.01%), males (51.00%), non-Hispanic Whites (70.15%), had some college degree (39.66%), had a total family annual income of ≥$75,000 (45.50%), and lived in the metropolitan commuting area (84.82%). After the declaration, significant proportions of participants also identified as SM (5.84%), reported being unemployed (40.67%), were not physically active (26.12%), and had CVD conditions (7.62%). Current e-cigarette use prevalence (4.79% vs. 8.63%) doubled from before to after the pandemic declaration. The prevalence also increased for no health insurance (8.21% vs. 9.43%), fair/poor general health status (13.30% vs. 14.55%), BMI (mean = 28.18 vs. 28.54), and moderate (6.44% vs. 8.75%) or severe (5.56% vs. 6.27%) anxiety/depression symptoms. The proportion of the participants living in the South increased from 36.67% to 38.61%.

### 3.2. Multivariable Logistic Regression Analyses

The results of the association of current e-cigarette use with the COVID-19 pandemic declaration, sexual identity, race/ethnicity, and the covariate are presented in Table 2 (Model 1). The multivariable logistic regression results showed that people aged 18–25 years (AOR = 7.58, 95% CI = 2.96, 19.44) had higher odds while those aged 65 years or older (AOR = 0.19, 95% CI = 0.06, 0.62) had lower odds of being current e-cigarette users compared to those aged 35–49 years. Compared to non-Hispanic Whites, non-Hispanic Blacks (AOR = 0.14, 95% CI = 0.03, 0.79) and Hispanics (AOR = 0.13, 95% CI = 0.05, 0.38) had lower odds of current e-cigarette use. The odds of current e-cigarette use were higher for SMs (AOR = 3.58, 95% CI = 1.32, 9.73) compared to heterosexuals. People who had college graduate or higher education, compared to High School graduates, had lower odds of current e-cigarette use (AOR = 0.33, 95% CI = 0.14, 0.79). Those who did not have health insurance, compared to their counterparts, had higher odds of current e-cigarette use (AOR = 3.23, 95% CI = 1.20, 8.66). The odds of e-cigarette use were approximately 13% higher after the pandemic declaration than before, but this was not statistically significant (AOR = 1.13, 95% CI = 0.56, 2.30).

Table 3 presents current e-cigarette use and its associated factors before and after the COVID-19 pandemic declaration, respectively. Compared to people aged 35–49 years, those aged 50–64 (AOR = 0.31, 95% CI = 0.10, 0.91) and 65 years or older (AOR = 0.01, 95% CI = 0.01, 0.09) had lower odds of current e-cigarette use before the pandemic declaration (Model 1), whereas the odds were higher for people aged 18–25 (AOR = 9.91, 95% CI = 3.29, 29.86) and 26–34 (AOR = 3.67, 95% CI = 1.08, 12.45) years after the declaration (Model 2). Those who lived in a micropolitan/small town/rural area, compared to those who lived in a metropolitan area, had lower odds of current e-cigarette use (AOR = 0.20, 95% CI = 0.04, 0.99) before the pandemic declaration. After the pandemic declaration (Model 2), non-Hispanic Blacks (AOR = 0.05, 95% CI = 0.01, 0.69) and Hispanics (AOR = 0.12, 95% CI = 0.03, 0.47) had lower odds of current e-cigarette use compared with non-Hispanic Whites. Compared to heterosexuals, SMs had higher odds of current e-cigarette use (AOR = 3.51, 95% CI = 1.10, 11.16) after the declaration. The odds of current e-cigarette use were lower for people who had a college graduate or higher education (AOR = 0.29, 95% CI = 0.09, 0.97) compared to High School graduates; and lower for individuals with a total family annual income of <$20,000 (AOR = 0.20, 95% CI = 0.05, 0.84) compared to those with $50,000 to <$75,000 after the declaration. BMI was associated with lower odds of e-cigarette use (AOR = 0.95, 95% CI = 0.90, 0.99) after the declaration. People who had no health insurance, compared to their counterparts, had higher odds of current e-cigarette use (AOR = 4.57, 95% CI = 1.42, 14.67) after the declaration. Having CVD conditions was associated with higher odds of e-cigarette use (AOR = 4.71, 95% CI = 1.88, 11.81) compared to not having CVD conditions after the declaration.

### 3.3. Between and within-Group Analyses

Figure 1 shows the differences in current e-cigarette use between and within race/ethnicity and the COVID-19 pandemic declaration. Non-Hispanic Asians (21.15%) had the highest probability of using e-cigarettes before the COVID-19 pandemic declaration, compared to non-Hispanic Whites (13.96%) and Hispanics (3.16%) after the declaration and non-Hispanic others (130.02%) and non-Hispanic Blacks (6.15%) before the declaration. The responses after the pandemic declaration showed a 4.53% (9.43–13.96%) increase in current e-cigarette use among non-Hispanic Whites and a 0.63% (2.53–3.16%) increase for Hispanics. The probability of current e-cigarette use among non-Hispanic Asians (3.05%), non-Hispanic Blacks (1.36%), and non-Hispanic others (5.53%) was higher before the pandemic declaration than after the declaration.

Figure 2 describes the differences in current e-cigarette use between and within-sexual identity and the COVID-19 pandemic declaration. In general, the probability of e-cigarette use was higher for SMs than for heterosexuals at both time points. Before the pandemic declaration, SMs had a significantly higher probability of using e-cigarettes compared to heterosexuals (21.20% vs. 7.25%). After the pandemic declaration, the probability of e-cigarette use significantly increased among heterosexuals (8.14%) but decreased among SMs (17.17%).

## 4. Discussion

Although studies on e-cigarette use continue to emerge in the U.S., to the best of our knowledge, this is the first study that used a nationally representative adult population to highlight disparities in e-cigarette use before and after the COVID-19 pandemic declaration. It was found that the prevalence of current e-cigarette use approximately doubled after the pandemic declaration from 4.79% to 8.63%. This increase is consistent with some earlier studies [38,39]. The prevalence of e-cigarette use also increased for U.S. adults with moderate and severe anxiety/depression symptoms, those without health insurance, and individuals with poor health during the pandemic. Evidence suggests that a wide range of pandemic-related factors such as increased stress from a potentially fatal disease, the possibility of a loss of employment, and feelings of insecurity, confinement, and boredom may serve as catalysts for increased use of e-cigarettes and tobacco products [39,40]. However, a recent study [32] reported lower odds of e-cigarette use but higher cigarette consumption post-lockdown compared to pre-lockdown. Noteworthy is the fact that Gonzalez et al. [32] used convenience samples of adults covering the most deprived counties in Central California [32,41]. Minorities and low-income smokers are more likely to believe e-cigarettes are more harmful than cigarettes and tend to have positive tobacco-related social norms [42].

Further, this study showed significantly elevated odds of e-cigarette use among young adults (18–34 years), SMs, adults without health insurance, and those with CVD conditions after the pandemic declaration. On the other hand, adults in the lowest income group, with the highest education levels, non-Hispanic Black people, and Hispanic people had significantly lower odds of e-cigarette use. E-cigarette use among young adults has always been of great concern [43] and our findings reinforce the urgent need to address this issue. Although earlier studies have reported declines in e-cigarette use among underage youth and young adults [23,30], it is still premature to speak of a declining trend [44]. Our report of increased usage in young adults (18–34 years) can be attributed to online proliferation [23,45] and unfair targeting of this population by the tobacco industry [46,47], especially during the pandemic.

Regarding race/ethnicity, we found that non-Hispanic Black people and Hispanic people were less likely to use e-cigarettes during the pandemic compared to non-Hispanic White people. Well-documented by studies even before the pandemic [42,48], this study’s findings confirm that racial/ethnic disparities persist among e-cigarette users. Thus, our findings suggest that the risk of e-cigarette use increased significantly among non-Hispanic White adults during the pandemic.

Additionally, those with the lowest income and highest level of educational attainment have lower odds of e-cigarette use, consistent with what other studies have reported [42,49]. Moreover, adults without health insurance reported higher odds of e-cigarette use after the pandemic declaration than those with health insurance. Similarly, Syamlal et al. [48] found U.S. adults without health insurance had almost twice the prevalence of e-cigarette use compared to those with health insurance. A possible reason for the decline in e-cigarette use among health insurance holders can be the tobacco rating, where insurance companies are allowed to charge tobacco users higher premiums [50]. Thus, while adults with health insurance are likely to quit e-cigarette usage due to increasing health insurance premiums, uninsured adults increased their use of e-cigarettes. As such, it is recommended that further studies should be performed to explore this association and assess the reasons for increased usage among uninsured adults during the pandemic. This can inform more comprehensive policies to reduce e-cigarette use in both insured and uninsured adults, as well as address some current controversies surrounding the taxation of e-cigarettes [45,51,52,53]. Moreover, this study found higher odds of e-cigarette use among SM individuals during this pandemic. Compared to their heterosexual counterparts, LGBTQ+ individuals are more likely to experience stigma and discrimination in addition to having a higher prevalence of chronic conditions associated with severe COVID-19 [54,55,56]. These can exacerbate stress [57] and increase e-cigarette use. Evidence also indicates that SMs are more likely to engage with tobacco-related messages on news and social media [58] and remain a prime target audience for the tobacco industry [59]. Further, individuals with CVD conditions reported higher odds of e-cigarette use compared to those without CVD conditions during the pandemic. This is very concerning because past studies have found associations between vaping and CVD conditions, including myocardial infarction [7,12,60]. Nonetheless, the most current evidence by Critcher et al. [61] suggests people who smoke and have suffered from a myocardial infarction or other negative effects of smoking may switch to e-cigarettes, generally perceived to be less harmful than tobacco cigarettes [62]. However, several uncertainties and temporality issues remain regarding overall cardiovascular health effects and e-cigarette use, warranting further investigation.

### Implications for Study Findings

The findings of our study indicate that although e-cigarette use has increased in the general US adult population during the pandemic, a substantial increase in use among subpopulations such as young adults, SMs, non-Hispanic Whites, individuals with CVD conditions, and those without health insurance may be driving this uptick. Even though priority populations were at increased risk of e-cigarette use even before the pandemic, the lockdown, with its concomitant restrictions, the online proliferation of e-cigarettes, and increased exposure to advertisements, may have exacerbated e-cigarette use and widened existing disparities. Thus, current and future policies and interventions to address the increased use of e-cigarettes and other substance use among U.S. adults should focus on these priority populations.

This study has some limitations. HINTS data are self-reported and may be subject to recall and social desirability bias, which could lead to underestimation. These data are also cross-sectional in nature, implying that no causal inferences can be made. The lack of self-reported data on e-cigarette use status is a further limitation of this study; however, due to the questions asked on the questionnaire, we feel the methodology for identifying e-cigarette use status described above is sound. We also could not examine nicotine consumption to determine their concentration because the dataset does not include this measure. Furthermore, we did not examine possible fears, behaviors, or COVID-19 disease/illness status in the context of the pandemic because the current HINTS data do not include COVID-19 behavioral-related measures. Despite these limitations, this is a nationally representative study with a large sample size that can be generalizable to the entire U.S. population.

## 5. Conclusions

This study has illuminated disparities in current e-cigarette use before and during the COVID-19 pandemic. Current e-cigarette use during the pandemic was high for SMs, people aged 18–34 years, non-Hispanic Whites, those with a CVD condition, those with no health insurance, and those with anxiety/depression symptoms. Although SMs had a higher probability of using e-cigarettes than their heterosexual counterparts, e-cigarette use increased slightly among heterosexuals during the pandemic. These findings highlight the need to further investigate the long-term effects of the pandemic on the use of substances such as e-cigarettes, especially between and within subpopulations.

## Figures and Tables

**Figure 1 ijerph-20-05649-f001:**
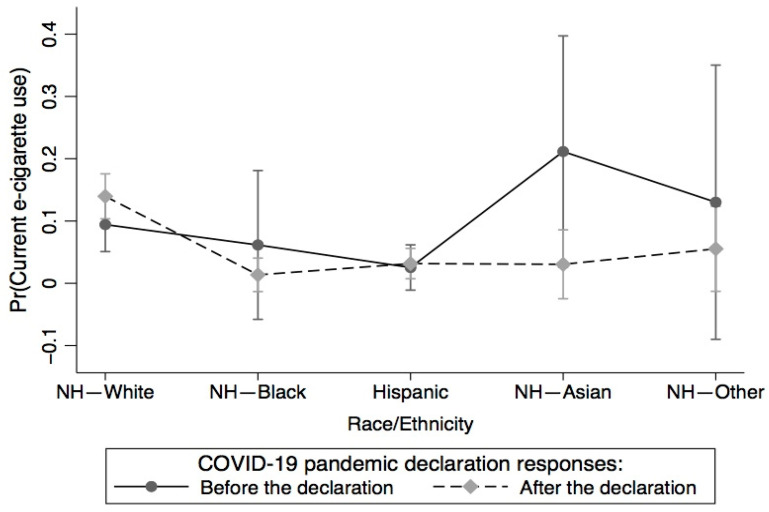
Differences in current e-cigarette use between and within race/ethnicity and COVID-19 pandemic declaration.

**Figure 2 ijerph-20-05649-f002:**
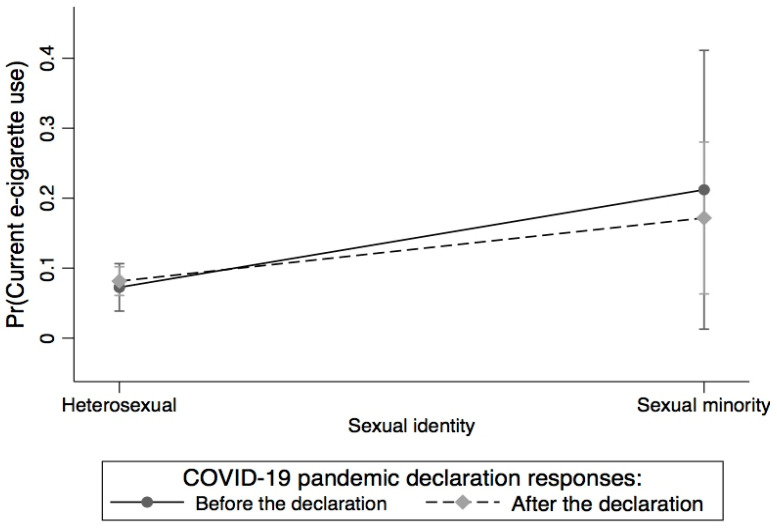
Differences in current e-cigarette use between and within sexual identity and COVID-19 pandemic declaration.

**Table 1 ijerph-20-05649-t001:** Weighted descriptive statistics of COVID-19 pandemic declaration across study participants (Weighted *N =* 253,815,197 and Unweighted *N =* 3865).

	Overall Sample	Responses before the COVID-19 Pandemic Declaration	Responses after the COVID-19 Pandemic Declaration
	*n* (%)	*n* (%)	*n* (%)
	3865 (100%)	1437 (35.31%)	2428 (64.69%)
Age			
18–25	147 (13.27)	41 (10.22)	106 (14.92)
26–34	337 (12.93)	110 (9.67)	227 (14.69)
35–49	703 (25.52)	212 (22.71)	491 (27.04)
50–64	1142 (27.72)	433 (33.01)	709 (24.87)
65 or older	1409 (20.56)	598 (24.39)	811 (18.48)
Sex			
Female	2204 (51.35)	804 (49.00)	1400 (52.63)
Male	1561 (48.65)	598 (51.00)	963 (47.37)
Race/ethnicity			
Non-Hispanic White	2133 (63.35)	904 (70.15)	1229 (59.63)
Non-Hispanic Black	481 (11.14)	135 (8.31)	346 (12.68)
Hispanic	596 (16.97)	170 (12.81)	426 (19.25)
Non-Hispanic Asian	161 (5.21)	51 (4.14)	110 (5.80)
Non-Hispanic other	119 (3.33)	49 (4.59)	70 (2.64)
Sexual identity			
Heterosexual	3402 (94.58)	2113 (95.36)	1289 (94.16)
Sexual and gender minority	163 (5.42)	56 (4.64)	107 (5.84)
Level of education completed			
Less than High School	273 (8.03)	90 (7.20)	183 (8.49)
High School graduate	705 (22.51)	251 (20.23)	454 (23.76)
Some college	1081 (39.18)	415 (39.66)	666 (38.92)
College graduate or higher	1663 (30.27)	643 (32.90)	1020 (28.84)
Total annual family income			
Less than $20,000	624 (15.15)	216 (14.15)	408 (15.60)
$20,000 to <$35,000	451 (11.46)	170 (10.85)	281 (11.80)
$35,000 to <$50,000	460 (12.67)	166 (12.03)	294 (13.01)
$50,000 to <$75,000	592 (18.26)	229 (17.47)	363 (18.70)
$75,000 or more	1321 (42.46)	514 (45.50)	807 (40.80)
Employment status			
Unemployed	1888 (40.89)	756 (41.31)	1132 (40.67)
Employed	1890 (59.11)	652 (58.70)	1238 (59.33)
Health insurance			
No	203 (9.00)	63 (8.21)	140 (9.43)
Yes	3604 (91.00)	1352 (91.79)	2252 (90.57)
Rural-urban commuting area			
Metropolitan	3387 (87.16)	1217 (84.82)	2170 (88.44)
Micropolitan/Small town/rural	478 (12.84)	220 (15.18)	258 (11.56)
U.S. Census region			
Northeast	581 (17.54)	225 (18.03)	356 (17.28)
Midwest	645 (20.83)	251 (21.81)	394 (20.30)
South	1728 (37.92)	634 (36.67)	1094 (38.61)
West	911 (23.71)	584 (23.50)	327 (23.82)
Number of adult household members			
At least two persons	2509 (79.00)	938 (80.50)	1571 (78.19)
Less than two persons	1356 (21.00)	499 (19.50)	857 (21.81)
General health status			
Excellent/very good/good	3192 (85.89)	1200 (86.70)	1992 (85.45)
Fair or poor	627 (14.11)	221 (13.30)	406 (14.55)
Current anxiety/depression status			
None	2670 (68.57)	1003 (70.03)	1667 (67.78)
Mild	629 (17.48)	246 (17.97)	383 (17.21)
Moderate	258 (7.93)	76 (6.44)	182 (8.75)
Severe	173 (6.02)	65 (5.56)	108 (6.27)
BMI (Mean, SD)	28.42 (6.71)	28.18 (6.43)	28.54 (6.84)
Moderate physical activity intensity			
None	1048 (27.11)	392 (28.93)	656 (26.12)
At least one day per week	2750 (72.89)	1729 (71.07)	1021 (73.88)
CVD Condition status			
None	3395 (91.88)	1247 (90.97)	2148 (92.38)
Yes	407 (8.12)	172 (9.03)	235 (7.62)
Current e-cigarette use status			
Never user	3314 (92.70)	1245 (95.21)	2069 (91.37)
Current user	114 (7.30)	40 (4.79)	74 (8.63)

Data Source: 2020 Health Information National Trends Survey (HINTS 5, Cycle 4). Weighted *N* = 253,815,197 and Unweighted *N* = 3865. BMI = Body Mass Index. SD = Standard deviation. CVD = Cardiovascular disease. Sexual minority (SM) = Homosexual or lesbian/gay/bisexual. Frequencies were not weighted while percentages and means with SD were weighted. Differences in total numbers in categories may be due to missing data.

**Table 2 ijerph-20-05649-t002:** Weighted multivariable logistic regression analysis of factors associated with current e-cigarette use (Unweighted *N =* 3428).

	Model 1	
	AOR	95% CI
Age		
35–49	Ref	
18–25	7.58 ***	(2.96, 19.44)
26–34	2.36	(0.93, 5.97)
50–64	0.55	(0.242, 1.25)
65+	0.19 **	(0.06, 0.62)
Sex		
Female	Ref	
Male	1.17	(0.63, 2.19)
Race/ethnicity		
Non-Hispanic White	Ref	
Non-Hispanic Black	0.14 *	(0.03, 0.79)
Hispanic	0.13 ***	(0.05, 0.38)
Non-Hispanic Asian	0.66	(0.17, 2.48)
Non-Hispanic other	0.45	(0.09, 2.36)
Sexual identity		
Heterosexual	Ref	
Sexual and gender minority	3.58 **	(1.32, 9.73)
Level of education completed		
High School graduate	Ref	
Less than High School	1.95	(0.51, 7.50)
Some college	0.89	(0.40, 1.98)
College graduate or higher	0.33 *	(0.14, 0.79)
Total family annual income		
$50,000 to <$75,000	Ref	
Less than $20,000	0.36	(0.12, 1.07)
$20,000 to <$35,000	0.87	(0.33, 2.30)
$35,000 to <$50,000	0.74	(0.26, 2.09)
$75,000 or more	0.67	(0.32, 1.38)
Employment status		
Employed	Ref	
Unemployed	0.8	(0.38, 1.68)
Health insurance		
Yes	Ref	
No	3.23 *	(1.20, 8.66)
Rural-urban commuting area		
Metropolitan	Ref	
Micropolitan/Small town/rural	1.12	(0.44, 2.88)
U.S. Census region		
South	Ref	
Northeast	0.82	(0.32, 2.09)
Midwest	0.78	(0.26, 2.31)
West	1.61	(0.71, 3.63)
Number of adult household members		
At least two persons	Ref	
Less than two persons	1.21	(0.69, 2.13)
General health status		
Excellent/very good/good	Ref	
Fair or poor	0.91	(0.35, 2.41)
Current anxiety/depression status		
None	Ref	
Mild	1.61	(0.69, 3.72)
Moderate	1.49	(0.60, 3.69)
Severe	1.66	(0.46, 5.94)
Body mass index (BMI)	0.97	(0.93, 1.02)
Moderate physical activity intensity		
At least one day per week	Ref	
None	1.08	(0.55, 2.11)
CVD Condition status		
None	Ref	
Yes	1.85	(0.66, 5.16)
COVID-19 pandemic declaration		
Responses before the declaration	Ref	
Responses after the declaration	1.13	(0.56, 2.30)

Data Source: 2020 Health Information National Trends Survey (HINTS 5, Cycle 4). Weighted *N* = 218,330,085 and Unweighted *N* = 3428. Model 1 = E-cigarette use + COVID-19 pandemic declaration + covariates. AOR = Adjusted odds ratio. 95% CI = 95% confidence interval. Ref = Reference group. CVD = cardiovascular disease. Sexual minority (SM) = Homosexual or lesbian/gay/bisexual. * *p* ≤ 0.05, ** *p* ≤ 0.01, *** *p* ≤ 0.001.

**Table 3 ijerph-20-05649-t003:** Weighted multivariable logistic regression analysis of factors associated with current e-cigarette use before and after the declaration of COVID-19 pandemic (Unweighted *N =* 3428).

	Model 2		Model 3	
	AOR	95% CI	AOR	95% CI
Age				
35–49	Ref		Ref	
18–25	4.63	(0.77, 27.74)	9.91 ***	(3.29, 29.86)
26–34	0.84	(0.20, 3.47)	3.67 *	(1.08, 12.45)
50–64	0.31 *	(0.10, 0.91)	0.58	(0.21, 1.64)
65+	0.01 ***	(0.01, 0.09)	0.24	(0.05, 1.22)
Sex				
Female	Ref		Ref	
Male	0.69	(0.25, 1.92)	1.59	(0.77, 3.29)
Race/ethnicity				
Non-Hispanic White	Ref		Ref	
Non-Hispanic Black	0.22	(0.05, 1.02)	0.05 *	(0.01, 0.69)
Hispanic	0.28	(0.06, 1.30)	0.12 **	(0.03, 0.47)
Non-Hispanic Asian	2.12	(0.48, 9.36)	0.15	(0.02, 1.49)
Non-Hispanic other	2.48	(0.28, 21.73)	0.21	(0.03, 1.44)
Sexual identity				
Heterosexual	Ref		Ref	
Sexual and gender minority	4.54	(0.88, 23.25)	3.51 *	(1.10, 11.16)
Level of education completed				
High School graduate	Ref		Ref	
Less than High School	2.04	(0.50, 8.37)	2.04	(0.43, 9.73)
Some college	0.73	(0.25, 2.14)	0.99	(0.41, 2.38)
College graduate or higher	0.62	(0.15, 2.57)	0.29 *	(0.09, 0.97)
Total family annual income				
$50,000 to <$75,000	Ref		Ref	
Less than $20,000	2.53	(0.52, 12.37)	0.20 *	(0.05, 0.84)
$20,000 to <$35,000	3.07	(0.59, 15.92)	0.5	(0.13, 1.82)
$35,000 to <$50,000	3.45	(0.47, 25.55)	0.42	(0.13, 1.33)
$75,000 or more	0.6	(0.17, 2.16)	0.64	(0.27, 1.49)
Employment status				
Employed	Ref		Ref	
Unemployed	0.41	(0.10, 1.65)	0.82	(0.36, 1.87)
Health insurance				
Yes	Ref		Ref	
No	1.27	(0.14, 11.21)	4.57 *	(1.42, 14.67)
Rural-urban commuting area				
Metropolitan	Ref		Ref	
Micropolitan/Small town/rural	0.20 *	(0.04, 0.99)	2.34	(0.85, 6.43)
US Census region				
South	Ref		Ref	
Northeast	1.06	(0.40, 2.80)	0.41	(0.12, 1.37)
Midwest	0.87	(0.18, 4.31)	0.67	(0.18, 2.43)
West	0.45	(0.09, 2.20)	2.08	(0.79, 5.46)
Number of adult household members				
At least two persons	Ref		Ref	
Less than two persons	0.35	(0.06, 1.92)	1.81	(0.87, 3.75)
General health status				
Excellent/very good/good	Ref		Ref	
Fair or poor	0.77	(0.07, 8.84)	0.86	(0.30, 2.48)
Current anxiety/depression status				
None	Ref		Ref	
Mild	1.36	(0.30, 6.12)	1.9	(0.66, 5.42)
Moderate	0.4	(0.05, 2.97)	2.16	(0.80, 5.85)
Severe	1.1	(0.12, 9.95)	2.27	(0.50, 10.38)
Body mass index (BMI)	0.99	(0.93, 1.07)	0.95 *	(0.90, 0.99)
Moderate physical activity intensity				
At least one day per week	Ref		Ref	
None	2.11	(0.84, 5.31)	0.85	(0.33, 2.18)
CVD Condition status				
None	Ref		Ref	
Yes	0.18	(0.01, 3.33)	4.71 ***	(1.88, 11.81)

Data Source: 2020 Health Information National Trends Survey (HINTS 5, Cycle 4). Model 2 (if before the declaration of COVID-19 pandemic) = e-cigarette use + covariates. Model 3 (if after the declaration of COVID-19 pandemic) = e-cigarette use + covariates. AOR = Adjusted odds ratio. 95% CI = 95% confidence interval. Ref = Reference group. Sexual minority (SM) = Homosexual or lesbian/gay/bisexual. * *p* ≤ 0.05, ** *p* ≤ 0.01, *** *p* ≤ 0.001.

## Data Availability

The data were extracted from the 2020 Health Information National Trends Survey (HINTS) de-identified public-use file. Detailed information about the methods and survey questions can be found in Health Information National Trends Survey 5 (HINTS 5) Cycle 4 [https://hints.cancer.gov/docs/methodologyreports/HINTS5_Cycle4_MethodologyReport.pdf (accessed 29 June 2021)].

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
