# Peer review of "Examining Disparities in Current E-Cigarette Use among U.S. Adults before and after the WHO Declaration of the COVID-19 Pandemic in March 2020"

_ijerph, 2023, doi:10.3390/ijerph20095649_

Round 1

Reviewer 1 Report

The publication by Mamudu et al. addresses an important and, above all, clinically relevant topic regarding epidmediology and the question of what impact the COVID-19 pandemic has on the user behavior of smokers and vapers. Due to the current development of a parallel increasing incidence of smokers, a parallel short presentation of the same data for classic nicotine consumption would be of great interest for understanding and, above all, for a scientific statement or theory/explanation approach. The methodological approach, but above all the introduction, is very clearly formulated and stringently comprehensible. Likewise, the conclusions of the interpretation can be derived one-to-one from the presented results, but would be meaningfully supplemented by the data of smoking consumption. Another major weakness of the work is the question of the COVID-19 pandemic impact itself. The survey, as it is presented at least in the methods section, does not address the concerns or possible fears in the context of the pandemic. Finally, it remains unclear whether the respondents themselves were ill with COVID-19. Thus, only the change itself is referred to. Unless otherwise possible, this would usefully be added in the area of limitations.

Author Response

Please see attached file for response to how we addressed issues that you raised in your earlier review.

Thank you,

Hadii M. Mamudu, PhD, MA, MPA FAHA

Reviewer 2 Report

This study assessed the prevalence of current e-cig use before and after the WHO COVID declaration and further investigated the prevalence among subgroups such as sex and race. Overall, the findings from this paper are interesting and important, but a few things need to be considered and improved.

1.      For introduction, since this study is assessing the disparities, I believe the introduction should weigh more on the disparities such as sex and race. Suggest expanding those details.

2.      Authors talked about the need for this study to investigate the e-cigarette use before and after the WHO COVID declaration and the disparities among adults. This study only included race and sex investigation. Why not also including the age since e-cigarette use as well as the reasons to use are varied among different age groups?

3.      For measures, I have a concern regarding the current e-cigarette use status. The way authors defined seems like there is lack of another category-lifetime users (not currently use but ever used), why not just using current users vs. non-current users? Also, for the two questions I don’t see the measurement for former use, did author use a third question? This is confusing to me. Suggest revising the relevant part.

4.      Why the reference group of age is 35-49?

5.      Authors included both gender and sex in the regression models. Did authors check if these variables are correlated? If so, please also include the description of the method in the analysis plan.

6.      For Tables 2&3, why not reporting them in one table?

7.      I appreciate the Figures 1&2 since it is clear to show me the changes of current e-cig use with statistical significance by the COVID declaration condition. I also suggest including a figure for age, as to compare the findings with the previous literature (especially for young adults).

8.      For the discussion, why authors discussing the selected covariates (e.g., did not include any of the anxiety findings). Instead, I would suggest being focus on the main findings such as the race and sex. Too much information for the covariates such as health insurance may be distracting.

9.      What are the implications for the study findings? Such as informing policies or interventions regarding the disparity. Maybe include those in the discussion can help reinforce the study aims.

Author Response

Thank you for the highly constructive feedback! Our response to your comments and suggestions are in the attached document.

Sincerely,
Hadii Mamudu, PhD, MA, MPA, FAHA

Round 2

Reviewer 1 Report

Dear authors,

Thank you for the revised version. After the revision, essential aspects have been addressed and, above all, weaknesses have been clearly identified in the publication, so that I would acknowledge the revision and speak out in favour of publication.